# FSN: FEATURE SHIFT NETWORK FOR LOAD-DOMAIN DOMAIN GENERALIZATION

## ABSTRACT

Conventional deep learning methods for fault detection often assume that the training and the testing sets share the same fault pattern spaces and domain spaces. However, some fault patterns are rare, and many real-world faults have not appeared in the training set. As a result, it's hard for the trained model to achieve desirable performance on the testing set. In this paper, we introduce a novel domain generalization, Load-Domain (LD) domain generalization, which is based on the analysis of the CWRU bearing dataset and its domain division method. For this scenario, we propose a feature shift model called FSN (Feature Shift Network). In the bearing dataset, domains are divided based on different operating conditions which have specific loads, so it's equivalent to load-based domain division. Moreover, the domain label corresponds to the actual load magnitude, making it unique as it contains physical information, which can boost detection accuracy on unknown domain beyond the training set. According to the knowledge above, FSN is trained for feature shift on adjacent source domains, and finally shifts target domain features into adjacent source domain feature space to achieve the purpose of domain generalization. Extensive experiments on CWRU demonstrate that FSN is better than the existed models in the LD domain generalization case. Furthermore, we have another test on MNIST, which also shows FSN can achieve the best performance.

## 1 INTRODUCTION

The complexity of machinery has been increasing over the past few years, and the traditional method of mechanical fault detecting cannot keep up with the level of automation of machinery. This paper focuses on exploring the mechanical faults in rotating machinery, and bearing failure detection is the primary focus of research on real-time rotating machinery detection.

Deep learning is undoubtedly the most widely used method for data-driven fault diagnosis. Deep neural network algorithms have achieved remarkable success in a variety of fields since the theory of Deep Learning was established(LeCun et al., 2015). Although it is typically assumed that the training set and the testing set share the same fault pattern space while training neural networks, this simply means that the feature data used for training contains all potential operating conditions. Despite our greatest efforts in dataset collecting, there are still undiscovered faults in actual industrial manufacturing. Models trained on existing operating conditions may not necessarily perform well in diagnosing faults in different operating conditions(Li et al., 2021). Therefore, to address the issue of real-world operating conditions that may not be included in the training set, this paper introduces domain generalization into the fault diagnosis model and makes improvement to commonly used adversarial domain generalization models.

Additionally, in order to achieve the goal of domain generalization, this paper introduces a special scenario for domain generalization called Load-Domain(LD) domain generalization and proposes a feature shift model FSN for it. In this scenario, features are shifted from the target domain into the source domain to achieve domain generalization. In conclusion,our contributions:

- Section 2 analyses the using traditional methods at home and abroad, the machine learning method, deep learning approach, and generalization method for fault diagnosis in the domain of the research status.

- Section 3 and 4 introduces a special domain generalization scenario called Load-Domain domain generalization, also known as LD domain generalization, and proposes a feature shift model (FSN) for this scenario.

- Section 5, experiments are conducted on the CWRU bearing dataset and compared with the classical fault diagnosis methods and domain generalization methods. The results demonstrate the superiority of the suggested method by analysing the benefits and drawbacks of the proposed FSN and the conventional methods. Additionally, experiments are performed on the MNIST dataset to evaluate several domain generalization methods. The comparison shows that FSN, which is the solution suggested in this paper, performs the best in the specific scenarios.

## 2  RELATED WORK

In genuine industrial production, fault diagnosis is a very real challenge. After more than 60 years of development, there are many mature methods for mechanical fault diagnosis thanks to the establishment of numerous international research facilities in the 1960s and afterward.

Domain generalization has become one of the current research hotspots in the field of deep learning in order to enhance the generalization capability of fault diagnosis models on datasets with unknown working conditions.

Traditional methods such as Cui et al. (2021) used the Seagull optimization algorithm (SOA) to determine the system parameters, input the signal into their proposed first-order multistable stochastic resonance system (CMSR), and then used Fourier transform on the output and obtained the fault diagnosis results. Video methods mainly include wavelet transform (Zhao et al., 2018; Peng & Chu, 2004), short-time Fourier transform (Wang et al., 2013; Karlsson et al., 2000) and other methods. Based on the Case Western Reserve bearing data set with Gaussian noise, Chen et al. (2022) organized its data points into grayscale images, and then used wavelet transform to process and input them into the neural network with Row Average Pooling (ARP) to obtain classification information, and achieved good anti-noise effect.

Traditional fault diagnosis methods require not only manual feature extraction, but also specific professional knowledge, so the methods are not adaptable and robust. Machine learning algorithms can automatically generate method models through iteration based on selected features, which greatly reduces labor costs . For example, Support Vector Machine (SVM) is a supervised learning method (Cortes & Vapnik, 1995). In classification, the dimension of the classification vector does not affect the performance of the model classification, which makes support vector machine better generalization in the domain of fault diagnosis, so SVM is one of the important tools for fault diagnosis. Fernández-Francos et al. (2013) input the vibration fault features extracted by the envelope spectrum into the support vector machine to classify the health data and fault data, and then perform fault diagnosis on the fault data to determine the fault mode. Aishwarya & Brisilla (2023) used various machine learning techniques (like SVM, K-nearest neighbors (k-NN),ML perceptron (MLP), Random Forest (RF), Decision Tree (DT), etc.)  to implement a fault detection strategy in the designed induction motors under variable load conditions.Dutta et al. (2023)presents a case study of a machinelearning (ML)-based computational technique for automatic fault detection in acascade pumping system based on variable frequency drive. These studies have improved the correct rate of fault diagnosis.

Although traditional machine learning methods can be trained automatically, features still need to be manually extracted, which may have the problem of incomplete feature selection and slow speed. Deep learning has completely solved this shortcoming, and has become the most popular and best tool in the domain of fault diagnosis. The first paper using convolutional neural networks(CNN) to detect bearing faults was published in 2016(Janssens et al., 2016), and since then, papers using CNN for bearing fault diagnosis have continued to appear. Guo et al. (2017); Chen et al. (2018); Qian et al. (2018) have done a lot of work in the domain of fault diagnosis, which provides new ideas for using CNN for fault diagnosis.

Recurrent neural Network (RNN) is suitable for processing sequence data, so it can be used for fault diagnosis. In 2015, Abed et al. (2015) applied recurrent neural networks to fault diagnosis. In

their paper, firstly, discrete wavelet transform was used to extract features, then the features were analyzed and selected, and finally RNN was used for classification.

Based on the idea of Adversarial, Xie & Zhang (2018) proposed a fault diagnosis method based on confrontation. In the model, an adversarial network was used to balance unbalanced signals. Experiments show that the classification performance of this method is better than other data balance methods in the case of imbalanced data sets.

Domain generalization belongs to transfer learning, which is a data-based method that mainly maps data belonging to different domains using a specific method, calculates the difference in distribution between data in different domains after mapping, and reduces the difference between domains after mapping by training the mapping method, so as to find a mapping method that minimizes the difference in the distribution of different domains to achieve domain generalization.

The domain generalization problem was first introduced as a machine learning problem by Blanchard et al. (2011) to solve cell classification problems in medicine, and has since been applied to transfer models between multiple domains. Muandet et al. (2013) used a complex nonlinear neural network to minimize the variance between the source and target domain features , and Li et al. (2018) made the source and target domain feature distributions more similar by minimizing the MMD distance between the source and target domain feature distributions and using adversarial learning. Without using distance methods, Motiian S achieved the domain generalization task using minimized contrastive loss (Motiian et al., 2017). In autopilot scenarios, the model does not change accordingly to real-world domain changes, which hinders the generalization of object detection across different real-world domains, and normalization perturbations are proposed to cope with classification domain generalizationFan et al. (2022). Bai et al. (2022) proposed a Temporal Domain Generalization with Drift-Aware Dynamic Neural Network to learn models under temporally changing data distributions and generalize to unseen data distributions following the trends of the change.

## 3   Problem Definition and Description

We start by introducing some of the symbols used to describe problems. The input space should be X, and the label space should be Y. Then a domain can be introduced as a joint distribution $P_{XY}$ in $X \times Y$. For a specific domain $P_{XY}$, $P_X$ presents marginal distribution in $X$, $P_{Y|X}$ denotes the posterior distribution of $Y$ given $X$, and $P_{X|Y}$ denotes the class-conditional distribution of $X$ given $Y$ .

In the scenario described in this section, we can obtain $K$ similar but different source domains $S$, where each source domain $S$ corresponds to a joint distribution $P_{XY}^{(k)}$. $\boldsymbol{S}$ ought to be the source domain space, i.e., $\boldsymbol{S} = \left\{ S_k = \left\{ \left( x^{(k)}, y^{(k)} \right) \right\} \right\}_{k=1}^{K}$.It should be noted that for $k \neq k^{'}$ and $k, k^{'} \in \{0, \ldots, K\}$ in the source domain, we have $P_{XY}^{(k)} \neq P_{XY}^{\left(k^{'}\right)}$. For the scenario described in this section, the goal is to get a predictive model $f : X \to Y$ on the source domain data, and to achieve the minimum prediction error on the unknown target domain data $T = \left\{ x^T \right\}$. The joint distribution corresponding to the target domain is denoted as $P_{XY}^{T}$, and similarly, $P_{XY}^{T} \neq P_{XY}^{(k)}$, $\forall k \in \{0, \ldots, K\}$.

The most important difference between the fault diagnosis scenario described in this paper and the common domain generalization situation is that for $\forall k \in \{0, \ldots, K\}$, the domain number $1 \cdots k$ has physical meaning. In the example shown in Figure 1, the domain transfer dataset used in Figure 1(a) is a collection of images in different styles, the domain labels contain no information only a tag. But for the CWRU bearing dataset shown in Figure 1(b), the domain labels in the bearing dataset represent the rotor load of the bearing during operation. Therefore, we can obtain some additional information from the domain number of the source domain data and the target domain data, and use it in the domain generalization. As a result, we introduces a new domain generalization scenario called Load-Domain(LD) domain generalization, to describe this special scenario.

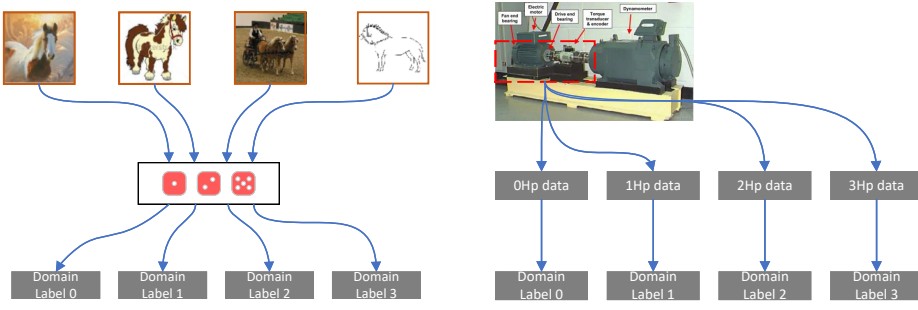

(a) Arbitrarily assigned domain labels       (b) Domain labels with actual physical meaning

Figure 1: Domain label with or without physical information. (a)For these four styles. The domain labels contain no information except that they can be used to distinguish between different domains. (b)For these four styles. The domain labels contain no information except that they can be used to distinguish between different domains.

## 4   FEATURE SHIFT NETWORKS

In last section, a special domain generalization scenario is introduced, and in this section, a feature shift model based on the inter-domain relationship will be proposed for this scenario. When the model is trained in the source domain, it can also achieve good fault diagnosis performance on unknown domains that have a specific relationship with the source domain. We call this model feature shift networks(FSN);

### 4.1   THEORETICAL ANALYSES

The design of FSN is based on two ideas, namely domain alignment idea and domain distribution regularity idea.

**Domain Alignment Conjecture.** The solution proposed in this section is based on the idea of domain alignment. For a domain, it can be modeled as a joint distribution $P(X, Y)$ (for ease of exposition in this section, we consider $P(X, Y)$ as $P_{XY}$). Then we can decompose this joint distribution as

$$
\begin{aligned}
P(X, Y) &= P(Y \mid X) P(X) & (1) \\
&= P(X \mid Y) P(Y) & (2)
\end{aligned}
$$

A common assumption in domain generalization is that shifts in the data distribution occur only at the edges $P(X)$, with the posterior $P(Y \mid X)$ remaining relatively stable. From the perspective of deep learning, if the marginal distributions of two domains can be aligned, then a predictive model trained on one domain can perform effectively as well on the other domain. In this case, the two domains can be viewed as having the same distribution. From the perspective of causal learning, if the assumption in the previous paragraph holds, then aligning the marginal distribution $P(X)$ is only valid when X is the cause of Y, because in this case, $P(Y \mid X)$ is not coupled with $P(X)$, and therefore $P(Y \mid X)$ can always remain stable when $P(X)$ changes. Of course, there is also a possibility that $Y$ may be the cause of $X$, in which case the shift of $P(X)$ will also affect $P(Y \mid X)$.

One of the theoretical bases of the solution proposed in this section is based on the above assumption, that is, $X$ is the cause of $Y$, $P(Y \mid X)$ is not coupled with $P(X)$, and therefore the data distribution shift only occurs on the margin$P(X)$. In this case, as long as the margin $P(X)$ is aligned, the model can be transferred from one distribution to another.

**Distribution Law Conjecture.** Besides the posterior stability idea mentioned above, the model proposed in this section is also based on another idea: When the domain labels correspond to physical meanings in reality, if there are two pairs of domain labels with identical internal relationships, then the internal domain feature distributions of the two pairs of domain data corresponding to these

two pairs of domain labels are also identical. More precisely, if we denote the $K$ domain labels as $D_1, D_2, \cdots, D_K$, which correspond to domains $P(1), P(2), \ldots, P^{(K)}$, if there exists a relationship mapping $F_d$, such that $F_d(D_{k'}) = D_k$, and $F_d(D_{k''}) = D_{k'}, k, k' \in \{0, \ldots, K\}$, then there exists a relationship $F_p$, when the relationship mapping between the domain features $f\left(P^{(k)}\right)$, $f\left(P^{(k')}\right)$ is $F_p$, that is, $F_p\left(f\left(P^{(k')}\right)\right) = f\left(P^{(k)}\right)$, then $F_p\left(f\left(P^{(k'')}\right)\right) = f\left(P^{(k')}\right)$.

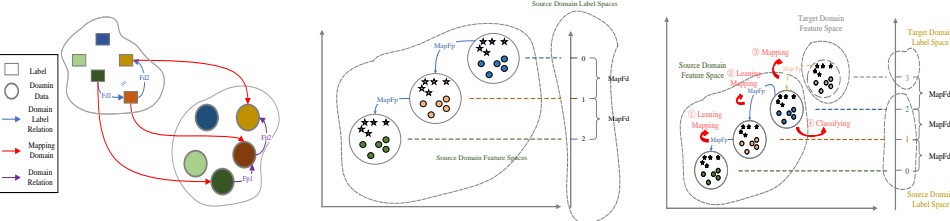

(a) Maping relations for domain-specific generalization

(b) An example of the label and domain feature space

(c) FSN design idea

Figure 2: Visualizations of mapping relations and design idea of FSN. (a) shows $F_p$ and $F_d$ how to map labels(features) to another. (b) clarifies domain labels that have a physical meaning and have simple relationships may have simple relationships between their domain feature distributions.(c) is the design idea of FSN.

As shown in Figure 2(a), within the domain labels, there is a mapping relationship $F_d$ that can map one label to another label, and then there is a mapping relation $F_p$ exist between the labels corresponding to these relationships, which can map the distribution of domain features to another domain features.

To illustrate the above more clearly in a diagram, let's take a simple example. For the source domain labels 0,1,2 of the CWRU dataset, it follows a linear distribution on the number line. We assume that the distribution of the domain features corresponding to the current domain labels is a kind of "linear" distribution in a certain dimension space. Figure 2(b) shows that these domain feature spaces visually appear to have a "linear" relationship. Because the source domain in the domain of tag actually corresponds to the bearing load in the running process, and is directly corresponding to the load size code size (not random number), so if domain label there is a relatively simple relations hip between (used in this dataset is linear relationship), so we have reason to believe that, the relationship between the feature Spaces of multiple domains corresponding to its labels can also be obtained by some means. If the mapping between these domains can be obtained, the data in the feature space of one domain can be completely mapped to the feature space of the other domain. Even if the mapping does not fit the real mapping well, as long as it can fit the mapping between the existing domain feature Spaces reasonably well, it is considered to be better than domain generalization without any additional information to guide it.

## 4.2 Transfer Methods between Different Distributions

One of the most popular approaches is Domain Alignment, which aligns the feature distributions across the source domains in order to better apply predictive models trained on the source domain to the unknown target domain. The reasoning behind this approach is as follows: if a feature learned on the source domain is insensitive to distributional offsets between the source domains, then for that feature, its is also relatively robust to distributional offsets on the unseen target domain. Currently, most of the methods related to domain generalization are developed based on the idea of domain alignment, and such methods are generally based on aligning the edge distribution $P(X)$, aligning the class-conditional distribution $P(X|Y)$, or aligning the posterior distribution $P(Y|X)$.

A general approach, in the scenarios discussed in this section, leads to a waste of information. Therefore, new domain generalization methods and processes are needed to retain and learn these regularities when we have seen the actual regularities between domains.

**New Approach** In the scenario discussed in this section, each source domain and target domain label correspond to an actual working condition of the bearing, and its numbers 0, 1, 2... correspond to

the actual physical meaning. Thus, an increase in the numbering of domain labels numbered 0, 1, and 2 represents an increase in their corresponding bearing loads, and with a linear relationship between the numbers, it is natural to expect that the feature spaces of their corresponding domains also have some relationship with each other. Extending this to unseen domains, if the target domain numbers also have actual physical meaning accordingly, then the target domain feature space should also have some kind of relationship with certain source domain feature spaces. The fact that the domain numbering adds physical meaning and that the target domain domain numbering is known is the biggest difference between the scenarios in this chapter and the domain generalization problem. Therefore, the traditional method of domain generalization through domain feature space alignment does not make any use of this added information, resulting in a waste of information. In order to make full use of the regular information between domains, this chapter proposes a new feature transformation model FSN, which is optimized for this special domain generalization scenario with additional information, and improves the practical effect of domain generalization.

## 4.3  DESIGN OF FSN

**Design Idea** FSN based on the assumption that features from different domains can be converted to each other as long as a pattern can be found. The design thought of its sources as shown in Figure  2(c).Since all tags in the label space of the source domain correspond to the actual working condition, and the load size in the working condition is the domain number, the number is practically meaningful in this scenario. There is a very simple mapping relationship between the numbers, showing a linear 0,1,2,3, which is evenly distributed on the number line shown on the right of the Figure  2(c). The domain numbers 0,1, and 2 are the source domain numbers, which correspond to the data

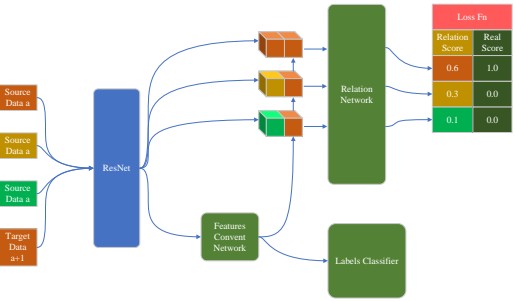

Figure 3: FSN generalization of the results

generated by the bearing under 0HP load, the data generated under 1HP load, and the data generated under 2HP load. These data belong to three domains, which together constitute all the source domain data. The domain number 3 is the data number of the target domain, which corresponds to the data generated by the bearing under 3HP load. These domain numbers correspond to the domain feature space on the left side of the figure. These domain feature Spaces are arranged in a row according to the domain label order, indicating that there is a certain pattern between these feature spaces.

**Network Structure** FSN consists of a feature extractor, a feature shift network, a relation classifier, and a label classifier. Both the feature extractor and the feature shift network are implemented by ResNet. The feature shift network receives the output feature of a feature extractor as input, outputs a feature of the same form, and expects this feature to be mapped to features in adjacent domains. The relation classifier is implemented in the same way as in Section 3 and is used for classification.

FSN structure is shown in Figure  3. During training, the input data are organized into the form of 1-9, i.e., 1 data of source domain a+1 and 9 data of source domain a, where these 9 data are derived from the 0 n classifications in domain a, and n denotes the number of classifications in the task. In the actual training, domain a may be domain 0, 1, and the corresponding domain a+1 is domain 1, 2. The data of domain a is changed into features after ResNet, and the extracted features of the data of domain a+1 have to go through a feature transformation network to get the transformed output, and it is expected that this output can be aligned with the features of the data with the same labels in domain a through the relation module, and the output is the relation of 1 Score. Eventually, the target domain 3 is taken as domain a+1, and the target domain 2 is domain a. At this point, the data in domain 3 is subjected to feature extraction as well as feature transformation, and the relationship scores are calculated with the features of each category in domain 2, and it is expected that the category with the highest relationship scores will be the real category of the data in the target domain.

## 5 Experiment

We conducted experiments for FSN on the CWRU dataset and the rotated MNIST dataset in LD domain generalization.

### 5.1 Evaluation in CWRU

**Setup** Datasets are used in this paper from western reserve university (CWRU) dataset of bearing, commonly referred to as CWRU bearing dataset. The dataset is an open-source dataset, made and released at case western reserve university, has been widely applied in the fault diagnosis and analysis.For more information refer to appendix A.

**Classical Model Contrast** In order to prove the effectiveness of FSN proposed in this section, FSN is compared with a variety of classical domain generalization methods.

The empirical risk minimization-based method (ERM) is first used for comparison. Although many methods have been proposed for domain generalization, which are intuitively reasonable and technically feasible, most of them can only achieve a small performance improvement (Zhu et al., 2022; Gulrajani & Lopez-Paz, 2020; Koh et al., 2021).So in this thesis, the ERM method as one of the methods compared with FSN .

In addition, DANN, which is adjusted for domain generalization, is also used as a comparison method. The domain adversarial neural network was actually proposed to solve the domain adaptation problem, and it was quickly applied to the domain generalization task similar to domain adaptation. Adaptive method is adjusted at the same time, a lot of areas, can also be used in domain generalization(Sicilia et al., 2023). This thesis DANN domain adaptive network adjustment, will source areas of these two classification networks adjustment for the source domain classification, generalization tasks make it can be used for domain, used in contrast experiment. The DANN adjusted for domain generalization will be directly denoted by DANN in the following.

On the basis of the above, in order to prove that FSN takes advantage of the features that domain labels have actual physical meaning, both single-source method and multi-source method experiments are carried out on the networks used in the experiment. Due to their high FSN need to train a can features to the nearest low domain shift in the domain of network, thus, FSN does not exist monophyletic method, it need at least two adjacent areas of tags. Again due to the need to put when generalization features into the adjacent areas of space, so FSN must use when double source domain experimental areas 1 and 2. For comparison, a l the models are trained on domain 1 and domain 2, and then the generalization test is performed on domain 3. In the multi-source domain experiment, all source domains 0-2 are used during training, and the domain generalization test is performed on the target domain 3. In addition, the relation classifier proposed in Section 3 is used to enhance the generalization ability during training.

Table 1 shows several ways of using multiple source domain in the domain of 3 generalization of results. Among them, the ERM said empirical risk minimization method, namely model training in the source domain, direct test on the target domain. As can be seen from the table, the multi-source domain model generalizes significantly better than the dual-source domain model. In the experiment of dual-source domain, FSN performs poorly, because the feature shift network is only trained on source domain 2→1, and the lack of domain makes the feature shift network unable to learn a good feature shift. In the multi-source domain

Table 1: Performance of the multi-source domain approaches

| No. | Model | Generalization accuracy | |
| --- | --- | --- | --- |
| | | FC | Relation |
| 0 | Double ERM | 75 | 71.4 |
| 1 | Double DANN | 76.3 | 78 |
| 2 | Double FSN | **75.2** | **72** |
| 3 | Multi ERM | 82.5 | 82.1 |
| 4 | Multi DANN | 83.3 | 82.6 |
| 5 | Multi FSN | **83** | **84.1** |

experiment, FSN achieves the best generalization performance. Explain FSN is very good use of the target domain number to carry in the domain of information, found the relationship between the features of adjacent areas.

In Table 1 at the same time shows each model use FC classifier's generalization effect and the relationship between generalization effect of classifier. We can see that the multi-source method is significantly better than the dual-source method, and FSN has the largest performance improvement when moving from dual-source to multi-source domains.

**Visual Analysis** Figure 4 shows that FSN is still 100% can be obtained in most of the failure mode of classification performance. For B021 and IR014 faults(more information rafer to Table A.1), the error of FSN is very serious, and only half of them is correctly classified. Among them, 27.02% of IR014 faults are identified as B021 faults, and 18.91% are identified as IR021 faults. For B021 fault, 40.54% of the data were identified as IR014 fault.

Figure 5(a) shows the t-SNE method to visualize the data. The failure mode names are represented by the corresponding numbers in Table A.1. When the test set data is in the original grayscale image state, many data of the same type are connected with each other, and have initially shown the features of locality. For example, most data of the same type are roughly connected together, and some data, such as fault No. 1 and fault No. 8, have been centrally distributed. Still mixed, however, other types of fault data, such as fault 3 scattered on the upper portion of the figure, fault 2 as a half ring scattered around the fault 1;fault 7,fault 0 are almost evenly mixed distribution. In the subsequent experiments, fault 1 and fault 8 show 100% classification accuracy when the model generalizes to the test set, which not only indicates that fault and fault 8 are easy to classify, but also indirectly proves the effectiveness of t-SNE plot in observing data distribution.

In addition, Figure 5(a) also shows that the clusters of fault classes 4 and 8 are close to each other, indicating that fault classes 4 and 8 May be difficult to separate in the classification. This is confirmed in the confusion matrix, where 20% of fault 4 is classified as fault 8. Nearly 25% of the classified results for fault 8 were incorrectly classified as fault 4, which was the only way to confuse fault 8.

The features output by the feature extractor of FSN are represented in Figure 5(b), and the feature output of the features in the figure after the feature shift network is shown in Figure 5(c). As you can see, to convert the area in front of the features and domain features after the shift, the intensity is roughly the same. However, in the experiment, use the converted classify the features of the test, the result is better than before the conversion feature. This indicates that the transformed feature space is closer to the source domain feature space.

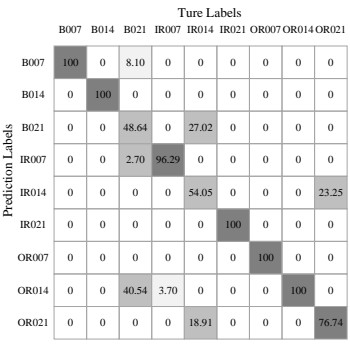

|  | Ture Labels | | | | | | | | |
|---|---|---|---|---|---|---|---|---|---|
|  | B007 | B014 | B021 | IR007 | IR014 | IR021 | OR007 | OR014 | OR021 |
| B007 | 100 | 0 | 8.10 | 0 | 0 | 0 | 0 | 0 | 0 |
| B014 | 0 | 100 | 0 | 0 | 0 | 0 | 0 | 0 | 0 |
| B021 | 0 | 0 | 48.64 | 0 | 27.02 | 0 | 0 | 0 | 0 |
| IR007 | 0 | 0 | 2.70 | 96.29 | 0 | 0 | 0 | 0 | 0 |
| IR014 | 0 | 0 | 0 | 0 | 54.05 | 0 | 0 | 0 | 23.25 |
| IR021 | 0 | 0 | 0 | 0 | 0 | 100 | 0 | 0 | 0 |
| OR007 | 0 | 0 | 0 | 0 | 0 | 0 | 100 | 0 | 0 |
| OR014 | 0 | 0 | 40.54 | 3.70 | 0 | 0 | 0 | 100 | 0 |
| OR021 | 0 | 0 | 0 | 0 | 18.91 | 0 | 0 | 0 | 76.74 |

(Prediction Labels)

Figure 4: FSN generalization of the results

Generally speaking, the features of the unseen target domain obtained by the trained FSN are more concentrated before and after the shift, which theoretically also shows that FSN method can have a good baseline domain generalization performance.

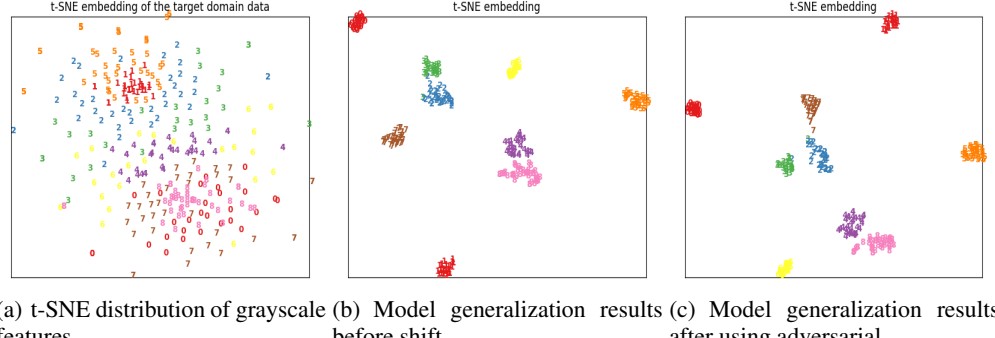

(a) t-SNE distribution of grayscale features

(b) Model generalization results before shift

(c) Model generalization results after using adversarial

Figure 5: t-SNE method to visualize the data. (a) shows distribution of grayscale features. (b)and(c) use t-SNE to show features performance model 5 before and after features shift.

## 5.2 Discuss on Other Datasets

We design and develop FSN for LD domain generalization, a novel domain generalization task. Compared with the ordinary domain generalization task, LD domain generalization has two special features: one is that its label number corresponds to some physical meaning, and the other is that the domain number of the target domain is known. We find that in addition to the CWRU bearing dataset used in this paper, there are some other datasets that also satisfy this characteristic. Therefore, we add a generalization experiment on the MNIST dataset for FSN to show its excellent performance in this specific scenario.

**Setup** In the domain generalization experiment conducted on the MNIST dataset, three domains in the training set 0-2 are used as the source domain, and the corresponding rotation angles are 0 degrees, 30 degrees, and 60 degrees. Wi l test 3 concentration domain as target domain, the corresponding rotation Angle of 90 degrees. The training process of the model was the same as the experimental process in 5.3.1. FSN was trained on the source domain, and then the model was used on the target domain to test the generalization effect.

Table 2: Generalization effect of each model on MNIST dataset

| Models | KDA | UB | DICA | SCA | MATE | ERM | DANN | DeepC | DeepN | CIDDG | FSN |
|---|---|---|---|---|---|---|---|---|---|---|---|
| Acc | 72.81 | 69.39 | 72.05 | 73.43 | 78.34 | 79.56 | 82.95 | 80.08 | 83.99 | 84 | 85.22 |

**Comparison with Classical Models** Table 2 shows FSN and the comparison of several other model. KDA, UB, DICA, SCA, MATE, ERM, DANN, DeepC, DeepN, CIDDG generalization accuracy of the model comes from Li et al. (2018) in the experiment, the experiment used the same as the experiment in this section in the process of the MNIST dataset, the standard of the same domain as we l as the same source domain and target domain, Therefore, the comparison can be made directly. The results show that our proposed FSN (which uses FC classifier for inference) achieves 1.5% higher generalization effect than CIDDG model in this scenario, which proves the effectiveness of FSN in the LD domain generalization task.

## 6 Conclusion

This study proposes LD domain generalization as a specific example of domain generalization, fully uses the particularity of the bearing dataset used for fault diagnosis, and implements FSN in this scenario. The domains in LD domain generalization correspond to the actual physical meaning. FSN learns the relationship between these domains, and finally shifts the data features in the target domain into the source domain for classification. Experiments reveal that FSN outperformed the comparison model's generalization performance by 1% to 2% on the CWRU dataset and reached 85.22% accuracy on the MNIST dataset.

However, the application of FSN is more restrictive. How to eliminate one or more of these limitations is also the goal of future research.

- Firstly, The model is only tested on the CWRU bearing dataset and the MNIST dataset. If the more dataset were tested, the more efficiency of model can be illustrated.

- Secondly, the proposed FSN requires at least three source domains, and the domain number is required to correspond to the actual physical information. In order to make other datasets meet this requirement, the datasets need to be selected carefully, and the appropriate domain partition method should be selected.

- Last, under the premise that the domain number corresponds to the actual physical information, the source domain numbering needs to be consecutive during the training process and can only be generalized relatively well from the target domain to the adjacent source domains. If the target and source domain numbers are not adjacent, then according to the idea on which the operation of this network is based, a better generalization cannot be achieved.

Therefore, the application of FSN model still is restrictive. How to eliminate one or more of these limitations is also the goal of future research.

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

## A  CWRU DATASET

Datasets are used in this paper from case western reserve university (CWRU) dataset of bearing, commonly referred to as CWRU bearing dataset. The dataset is an open-source dataset, made and released at case western reserve university, has been widely applied in the fault diagnosis and analysis. In order to facilitate the experiment, this paper takes the load as a field (0 3HP in total 4 fields). Under each load, three fault sizes are selected for the three fault locations: sphere, inner race, and 6 o 'clock of outer race. Therefore, there are 9 fault modes under each load. The data acquisition frequency of 12,000 Hz was chosen because some data was missing.

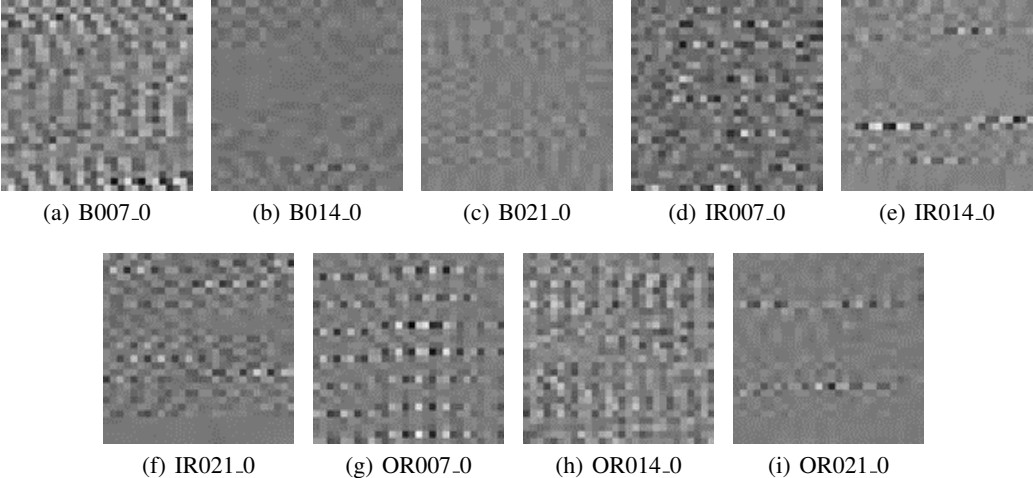

| (a) B007_0 | (b) B014_0 | (c) B021_0 | (d) IR007_0 | (e) IR014_0 |

| (f) IR021_0 | (g) OR007_0 | (h) OR014_0 | (i) OR021_0 |

Figure A.1: The wavelet time-frequency diagram of 10 fault modes under working condition 0.

Table A.1: CWRU bearing dataset

| Fault number | Fault location | Fault size | Lodd horsepower(HP) | | | |
|---|---|---|---|---|---|---|
| | | | 0 | 1 | 2 | 3 |
| 0 | | 0.007 | B007_0 | B007_1 | B007_2 | B007_3 |
| 1 | Ball | 0.014 | B014_0 | B014_1 | B014_2 | B014_3 |
| 2 | | 0.021 | B021_0 | B021_1 | B021_2 | B021_3 |
| 3 | | 0.007 | IR007_0 | IR007_1 | IR007_2 | IR007_3 |
| 4 | Inner race | 0.014 | IR014_0 | IR014_1 | IR014_2 | IR014_3 |
| 5 | | 0.021 | IR021_0 | IR021_1 | IR021_2 | IR021_3 |
| 6 | | 0.007 | IR007_0 | IR007_1 | IR007_2 | IR007_3 |
| 7 | 6 o'clock of outer race | 0.014 | IR014_0 | IR014_1 | IR014_2 | IR014_3 |
| 8 | | 0.021 | IR021_0 | IR021_1 | IR021_2 | IR021_3 |

Each load (field) below has 9 kinds of failure mode, by different fault location and size. When numbering each load condition, for each load number, the value directly corresponds to the horsepower value of the load (0 to 3), and for the fault mode number, the details are given in Table A.1.

For bearing signal processing, normalized firstly, then the size of 784, sliding step length is 200 overlap sampling window., and will be one dimensional signal is converted to gray image. In this case, there are about 600 grayscale images of 28*28 size in each fault mode of each domain. Taking the fault image under domain 0 data generated by condition 0 as an example, the corresponding grayscale images for each fault mode are shown in Figure A.1.

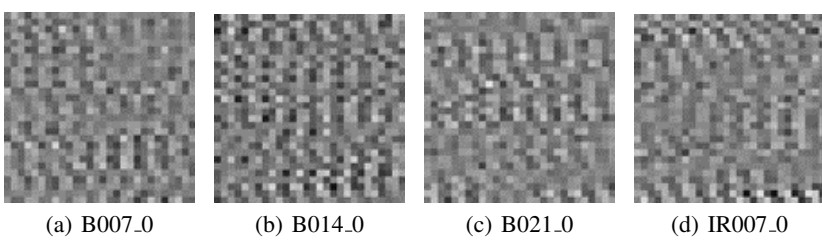

| (a) B007_0 | (b) B014_0 | (c) B021_0 | (d) IR007_0 |

Figure A.2: Cases 0 of 10 kinds of failure mode of the wavelet time-frequency diagrams.

It can be seen that the grayscale images of different fault modes under the same working condition are quite different. Taking the first fault mode B007_0 under four working conditions as an example, the grayscale diagram is shown in Figure A.2.

It can be seen that even the grayscale images of the same fault mode have different features if their working conditions are different.

## B  TRAINING DETAILS

In the FSN experiment, taking bearing load as the domain, the model is trained in multiple source domains to classify nine fault modes, and then the model is applied to the target domain data for generalization test. The number of samples is still about 600 samples for each fault mode, and each domain contains data for 9 fault modes.

The data with load of 0-2HP is taken as the three source domains with field number 0-2, and the data with load of 3HP corresponding to field number 3 is taken as the target domain, that is, the test set. The Epoch was set to 20, and 1200 iterations were carried out in each Epoch. After 50 iterations, model will be tested on the test set, to obtain the real-time effect of generalization. In order to reduce the randomness of the final results, a list is used to store the accuracy of the last 30 tests, and the average of the last 30 tests is given.

Before experiment, first describe the training process feature shift model. The FSN model is trained in stages to achieve better performance, and the training process of each stage is shown in Algorithms 1 2 3.

---

**Algorithm 1** Trains the feature extractor and the relation classifier

**Input:** $x_0, x_1, \ldots, x_9$ //$x_1, \ldots, x_9$ are grayscale image with labels 0-8 from the source domain
**Output:** $\theta_f$, $\theta_r$ // are the parameters of feature extractor and relation module respectively
 1: $x_0, x_1, \ldots, x_9$ = get()
 2: $feature_0 = G_f(x_0)$
 3: **for** $i = 1 \rightarrow 9$ **do**
 4:     $feature_i = G_f(x_i)$
 5:     $feature_{i,0} = concatenate(feature_i, feature_0)$
 6:     $relation\_score = G_r(feature_{i,0})$ //$G_r$ represents relation module
 7:     **if** $label(x_0)! = label(x_i)$ **then**
 8:         $loss+ = loss\_func(relation\_score, 0)$
 9:     **else**
10:         $loss+ = loss\_func(relation\_score, 1)$
11:     **end if**
12: **end for**

---

Algorithm 1 is the first stage in the three stages of FSN training, which uses the method of relation classifier for classification training to obtain the parameters of feature extractor and relation module.

Algorithms 2, the second stage of FSN training, add a fully connected classifier to the model and train it while updating the parameters of all components.

Algorithm refalg3 is the third stage of FSN training, which mainly trains the feature transformation network. It should be noted that in Algorithm 3, $x_1, \ldots, x_9$ not only correspond to labels 0 8, but also their domain numbers are always 1 smaller than $x_0$. By this setting, we can promote the alignment of features from high domains with features from low domains with the same label during training.

In general, FSN hopes to achieve better results through the above staged training process. In the training process, the network parameters trained in the previous stages will be used as the input for the subsequent stages, indicating that the network used in the subsequent stages is trained by the previous stages. By learning the transformation of domain features $1 \overset{shift}{\rightarrow} 0$ and $2 \overset{shift}{\rightarrow} 1$ on the source domain, we expect to learn a feature transformation network that can achieve the transformation of target domain 3 features to source domain 2 features $3 \overset{shift}{\rightarrow} 2$, thereby achieving the purpose of generalization.

---

**Algorithm 2** Train fully connected classifier

---

**Input:** $x_0, x_1, \ldots, x_9, \theta_f, \theta_r$
**Output:** $\theta_f, \theta_r, \theta_c$ //$\theta_c$is the parameters of the fully connected classifier
1: $x_0, x_1, \ldots, x_9$ = get()
2: $feature_0 = G_f(x_0)$
3: $loss+ = loss\_func(G_c(feature_0), label(x_0))$
4: **for** $i = 1 \to 9$ **do**
5:    $feature_i = G_f(x_i)$
6:    $feature_{i,0} = concatenate(feature_i, feature_0)$
7:    $relation\_score = G_r(feature_{i,0})$
8:    **if** $label(x_0)! = label(x_i)$ **then**
9:      $loss+ = loss\_func(relation\_score, 0)$
10:   **else**
11:      $loss+ = loss\_func(relation\_score, 1)$
12:   **end if**
13: **end for**

---

**Algorithm 3** Train fully connected classifier

---

**Input:** $x_0, x_1, \ldots, x_9, \theta_f, \theta_r, \theta_c$
**Output:** $\theta_f, \theta_r, \theta_c, \theta_s$ //$\theta_s$is the parameter of FSN
1: $x_0, x_1, \ldots, x_9 = get\_different\_domain()$ //$x_0$from domain n, $x_1, \ldots, x_9$ doman n-1
2: $feature_0 = G_f(x_0)$
3: $feature_{shift} = G_s(feature_0)$ //$G_s$is FSN
4: **for** $i = 1 \to 9$ **do**
5:    $feature_i = G_f(x_i)$
6:    $feature_{i,shift} = concatenate(feature_i, feature_s hift)$
7:    $relation\_score = G_r(feature_{i,shift})$
8:    **if** $label(x_0)! = label(x_i)$ **then**
9:      $loss+ = loss\_func(relation\_score, 0)$
10:   **else**
11:      $loss+ = loss\_func(relation\_score, 1)$
12:   **end if**
13: **end for**
14: $loss+ = loss\_func(G_c(feature_{i,shift}), label(x_0))$

---

