# OpenReview forum: "FSN: Feature Shift Network for Load-Domain Domain Generalization"
_ICLR.cc/2024/Conference — ICLR 2024 Conference Withdrawn Submission_

### Official Review · Reviewer_hzsf · 2023-10-25

**Soundness:** 2 fair
**Presentation:** 4 excellent
**Contribution:** 2 fair
**Rating:** 6
**Confidence:** 4

**Summary:**

The paper addresses a challenge in deep learning-based fault detection, where real-world faults may not always appear in the training set. This limitation makes it difficult for conventional models to generalize well to unseen fault patterns. To address this, the authors introduce a domain generalization method, Load-Domain (LD) domain generalization, specifically designed based on the CWRU bearing dataset. In this method, the domains are divided based on different operating conditions that have specific loads, which correspond to the actual load magnitude. This inclusion of physical information helps enhance the model's accuracy for unknown domains. The authors propose the Feature Shift Network (FSN), which is trained to shift features between adjacent source domains and the target domain for better generalization. The effectiveness of FSN is shown through experiments on both the CWRU bearing dataset and the MNIST dataset, where FSN outperforms existing methods.

**Strengths:**

+ The authors tackle a crucial issue in fault detection where conventional models may not perform well on unseen fault patterns.
+ This novel domain generalization approach, based on real physical properties (load magnitude), can potentially be more representative and robust than abstract or purely data-driven domain divisions.
+ The model's applicability on both the CWRU bearing dataset and the MNIST dataset suggests it is versatile and not limited to one type of data.
+ The authors compare their model with classical fault diagnosis methods, showing its superiority in specific scenarios.
+ The paper appears to have a well-structured format, with sections dedicated to reviewing the current state of research, introducing their novel domain generalization method, and discussing experimental results.

**Weaknesses:**

- The assumption that domain label corresponds to actual load magnitude might not hold for all real-world scenarios. It may be beneficial to test scenarios where this is not the case.
- Introducing domain-specific information like load magnitude could risk overfitting to specific domain characteristics. The generalization capability of the model in truly unseen domains is a concern.

**Questions:**

- How does the Load-Domain (LD) domain generalization approach compare with other domain generalization methods that don't rely on physical information like load magnitude?
- Are there potential scenarios where the assumption of domain labels corresponding to actual load magnitude may not hold? How would FSN perform under such conditions?

---

### Official Review · Reviewer_gaFV · 2023-10-28

**Soundness:** 1 poor
**Presentation:** 2 fair
**Contribution:** 1 poor
**Rating:** 3
**Confidence:** 5

**Summary:**

The paper proposes a feature shift network (FSN) for a new domain generalization task called load-domain (LD) generalization based on analyzing the CWRU-bearing dataset. The key idea is to leverage domain labels to shift target features into adjacent source domains. The method is evaluated on the CWRU and rotated MNIST datasets by comparing them to existing domain generalization techniques.

**Strengths:**

The application of domain generalization on the problem of fault detection is novel.

**Weaknesses:**

The paper lacks novelty as feature shift is explored in prior work. The theoretical analysis relies on unproven conjectures and lacks rigor. More comprehensive empirical evaluation on realistic datasets and comparisons to recent benchmarks are needed to demonstrate effectiveness.

**Questions:**

The novelty of the proposed feature shift network (FSN) is questionable given prior work on feature shift for domain generalization. The current paper does not sufficiently differentiate FSN from these existing methods.

The paper also lacks theoretical analysis regarding the generalization abilities of FSN.

 Empirically, evaluation is limited to two small datasets, which cannot sufficiently demonstrate effectiveness and robustness for the load-domain generalization task. More extensive testing on diverse realistic datasets are needed.

---

### Official Review · Reviewer_nQ8N · 2023-10-30

**Soundness:** 2 fair
**Presentation:** 2 fair
**Contribution:** 2 fair
**Rating:** 3
**Confidence:** 3

**Summary:**

In this paper, the authors introduce a new Load-Domain (LD) domain generalization setting, where the domain label corresponds to actual load magnitude. To serve the scenario, the authors propose a feature shift model (FSN) to learn feature mapping between adjacent domains according to the physical meaning in domain labels. Experiments are carried out on CWRU bearing dataset and rotated MINST datasets to showcase the performance.

**Strengths:**

1. The idea of employing the physical meaning of domain labels to achieve generalization on consecutive domains makes sense to me.
2. The new setting of LD domain generalization also seems suitable for the benchmark of CWRU dataset, where the physical meaning of domain labels can be clearly defined.

**Weaknesses:**

1. The paper’s writing and formatting are not good enough, making it sometimes hard to read and understand. There exist many formatting errors (e.g., the use of spaces in Sections 1 and 2, the math symbol consistency in Section 4.1: P(1) and P^{(K)}). The caption of Figure 1 is duplicated. The abbreviation in Paragraph 3, Section 2 might be RAP. In the second line of paragraph in section “Distribution Law Conjecture”, F_p should be the the mapping relation between features, not labels.
2. Besides the ones from 1., the Experiment part (e.g., paragraph 4 in “Classical Model Contrast” part) is hard to understand. Bad formatting and discontinuous sentences make it unreadable to me.
3. Key references are absent. In the first paragraph of Section 4.2, methods aligning the three types of distributions should be cited to justify the categorization. The compared methods in “Comparison with Classical Models” should also be referred to, and the Li et al. (2018) reference in this paragraph is wrongly cited, this paper should be “Deep Domain Generalization via Conditional Invariant Adversarial Networks”.
4. As to the experimental results, the metric of Table 1 is not clearly stated. Also, the experimental details like backbone choice, learning rate, optimization schedules are not provided.
5. The setting is still too limited. The current method and experiments only focus on generalizing from highly relevant and sequential source domains to a target domain close to the last seen source domain. The performance of the proposed method should be further evaluated on broader settings where the relationship of source and target domains is not fixed. Also, the setting seems much relevant to that of Continuous Domain Generalization, which should be discussed.

**Questions:**

Apart from those in weakness, the authors are encouraged to experiment on larger domain generalization benchmarks. Moreover, the compared methods are not recent enough, therefore more experiments should be added to provide a more comprehensive comparison.

---

### Official Review · Reviewer_u8DP · 2023-11-01

**Soundness:** 2 fair
**Presentation:** 2 fair
**Contribution:** 2 fair
**Rating:** 3
**Confidence:** 4

**Summary:**

The paper introduces a novel approach called Load-Domain (LD) domain generalization for fault detection in situations where the training and testing sets have different fault pattern spaces and domain spaces. The authors propose a feature shift model called FSN (Feature Shift Network) specifically designed for LD domain generalization. The model is trained on adjacent source domains to learn feature shifts and then applies these shifts to target domain features, enabling generalization beyond the training set. The approach is validated through extensive experiments on the CWRU bearing dataset and the Rotation MNIST dataset, demonstrating superior performance compared to existing models in LD domain generalization scenarios.

**Strengths:**

1. The topic of Domain Generalization is highly relevant and of significant interest to the research community. Furthermore, the paper addresses the important aspect of leveraging domain information during the training process, which has gained increased attention in recent times.
2. The paper is well-written and easily comprehensible, effectively conveying its ideas and findings to the readers.

**Weaknesses:**

1. The LD domain generalization problem setting addressed in the paper is acknowledged as a highly specialized case, which limits its broader contribution to the field.
2. The experimental results presented in the paper lack persuasiveness. Firstly, the dataset used is small and may not accurately represent real-world scenarios. Additionally, the chosen baselines are outdated, primarily predating 2018, despite the emergence of numerous domain generalization methods since then. It is recommended to refer to recent surveys for a comprehensive overview of the latest approaches, e.g., [1].
3. While the main idea of the paper is generally understandable, there are instances where sentences may create misunderstandings, and insufficient definitions of the problem and task settings are found throughout the paper. Notably, the captions for Figure 1 may inaccurately describe Figure 1(b), potentially causing confusion among readers.

[1] Domain Generalization: A Survey

**Questions:**

Please review the weakness section, and kindly correct any misunderstandings that may exist in my assessment.

---

### Official Review · Reviewer_xHmt · 2023-11-01

**Soundness:** 2 fair
**Presentation:** 1 poor
**Contribution:** 2 fair
**Rating:** 3
**Confidence:** 4

**Summary:**

This paper introduces a special domain generalization scenario termed Load-Domain domain generalization and proposes a new model, the Feature Shift Network (FSN), tailored for this scenario. The authors conduct experiments on the CWRU bearing dataset and the MNIST dataset, comparing FSN with classical fault diagnosis methods and other domain generalization methods.

**Strengths:**

The experimental results show good performance of FSN in certain scenarios and hence its potential for practical applications in fault diagnosis and domain generalization.

**Weaknesses:**

1. The writing of the paper should be substantially improved. It reads like bad machine translation and has a lot of grammatical and terminological errors.
2. The theoretical foundation of the FSN model could be explained in more detail. The model architecture shown in Fig. 3 requires further clarification and motivation.
3. Experiments are only conducted on two datasets, with limited baseline methods for comparison. It is unclear how the proposed method performs on other datasets that also have "linear" domain labels.
4. The accuracy values do not have confidence intervals.

**Questions:**

See above.

---

### Official Review · Reviewer_NDUe · 2023-11-02

**Soundness:** 2 fair
**Presentation:** 2 fair
**Contribution:** 2 fair
**Rating:** 3
**Confidence:** 4

**Summary:**

The paper aims to address the multi-source domain generalization problem by proposing a feature shift network (FSN). Distinct from traditional domain generalization, the problem of interest has additional information of inter-domain linear relations in the form of domain labels. By taking advantage of such information explicitly in the proposed model, superior performance can be achieved.

**Strengths:**

--The paper attempts to addresses the domain generalisation problem derived from a real-world application. The problem itself is somewhat novel and has not been extensively studied and hence solving such a problem is of great significance.

**Weaknesses:**

--The introduction section lacks essential information of problem definition, description of methods and brief experimental results. This makes it less readable to the readers. For example, it is not clear what the "Load-Domain" means and how the feature shift model handles the problem.

--The section of related work is not well organized. More focus should have been put on the most closely related works (i.e. domain generalization in fault detection problems) rather than a broad review of fault detection methods. In addition, the relations between existing works and this work should also be discussed.

--In table 1, Multi FSN does not perform the best as Multi DANN has a result of 83.3.

--The authors fail to compare with SOTA domain generalization methods.

--There exist language issues/typos/notation inconsistency in the manuscript. E.g., "P(1), P(2), ..., P^{(K)}"; "edge distribution" should be "marginal distribution"; "1 data of source domain a+1 and 9 data..."; "This thesis DANN domain adaptive network..."; "the ERM said empirical ...";

**Questions:**

1. What is the loss function of the network?
2. What is the input (images?) and the output of the model?

---

### Official Review · Reviewer_4bLC · 2023-11-04

**Soundness:** 2 fair
**Presentation:** 2 fair
**Contribution:** 2 fair
**Rating:** 5
**Confidence:** 3

**Summary:**

Traditional deep learning methods for fault detection usually assume that the training set and the test set share the same fault mode space and domain space. Based on the analysis of CWRU bearing data set and its domain division method, this manuscript proposes a Feature Shift model called FSN (Feature Shift Network) to improve the detection accuracy of unknown domain, which can divide domains according to different operating conditions with specific loads, and take advantage of the physical significance of domain labels.

**Strengths:**

1.This manuscript proposes the idea of "exploitability" of domain-related information, and it may be a point worth exploring further.
2.This manuscript is written in a standard and clear hierarchy,  and the structure is easy to follow.

**Weaknesses:**

1.In this manuscript, a parameter with physical significance is used as a domain label, and then the related features of the domain are used to assist classification. But now that labels have physical meaning, what happens if they are input directly to the network with other data? We didn't see the related comparison experiments. So it is not  convincing.

2.Few comparison experiments are conducted.

3.The detailed design of the model, including the loss function, is not explained in sufficient detail.

**Questions:**

1.Why label a domain with a piece of information that can be numerically and physically meaningful, and how is that different from feeding it directly into a neural network?

---

### Official Review · Reviewer_2Hv6 · 2023-11-08

**Soundness:** 2 fair
**Presentation:** 1 poor
**Contribution:** 1 poor
**Rating:** 3
**Confidence:** 3

**Summary:**

This paper targets a domain generalization problem for fault diagnosis of the bearing dataset and proposes a new model, called the feature shift network (FSN), to adjust the features between source and target domains. The numerical experiment using CWRU and MNIST datasets is conducted to evaluate the effectiveness of the proposed FSN.

**Strengths:**

- The motivation to exploit the additional information of the problem by assuming the specific task of fault diagnosis in bearing datasets is good.
- The domain generalization problem treated in this paper is important.

**Weaknesses:**

- The paper is not well written and has a lot of unclear points. For example, the detailed setting, such as loss function and calculation of each model component, is omitted. It is hard to understand the technical novelty and advantages of the proposed method.
- I cannot find the formal definition of the load-domain (LD) generalization problem treated in this paper.
- In Table 1, the performance gain of the proposed FSN variants is marginal.
- The motivation of the evaluation using the MNIST dataset is unclear.

**Questions:**

- What does the "Relation" mean in Table 1?
- How does the proposed FSN exploit the physical meaning of domain labels for model training?
- Is the target domain data accessible in the load-domain generalization? In general, the target domain data is not accessible in domain generalization.